# 5-Hydroxytryptamine Enhances the Pacemaker Activity of Interstitial Cells of Cajal in Mouse Colon

**DOI:** 10.3390/ijms25073997

**Published:** 2024-04-03

**Authors:** Xingyou Huang, Seok Choi, Wenhao Wu, Pawan Kumar Shahi, Jun Hyung Lee, Chansik Hong, Jae Yeoul Jun

**Affiliations:** 1Department of Physiology, College of Medicine, Chosun University, Gwangju 61452, Republic of Korea; huangxingyou93@gmail.com (X.H.); choiseok@chosun.ac.kr (S.C.); 17702903275@163.com (W.W.); pshahi@wisc.edu (P.K.S.); cshong@chosun.ac.kr (C.H.); 2Department of Internal Medicine, College of Medicine, Chosun University, Gwangju 61452, Republic of Korea; pp3614@chosun.ac.kr

**Keywords:** interstitial cells of Cajal, colon, pacemaker activity, hyperpolarization-activated cyclic nucleotide-gated channel

## Abstract

We examined the localization of the 5-hydroxytryptamine (5-HT) receptor and its effects on mouse colonic interstitial cells of Cajal (ICCs) using electrophysiological techniques. Treatment with 5-HT increased the pacemaker activity in colonic ICCs with depolarization of membrane potentials in a dose-dependent manner. Hyperpolarization-activated cyclic nucleotide-gated (HCN) channel blockers blocked pacemaker activity and 5-HT-induced effects. Moreover, an adenylate cyclase inhibitor inhibited 5-HT-induced effects, and cell-permeable 8-bromo-cAMP increased the pacemaker activity. Various agonists of the 5-HT receptor subtype were working in colonic ICCs, including the 5-HT_4_ receptor. In small intestinal ICCs, 5-HT depolarized the membrane potentials transiently. Adenylate cyclase inhibitors or HCN blockers did not show any influence on 5-HT-induced effects. Anoctamin-1 (ANO1) or T-type Ca^2+^ channel blockers inhibited the pacemaker activity of colonic ICCs and blocked 5-HT-induced effects. A tyrosine protein kinase inhibitor inhibited pacemaker activity in colonic ICCs under controlled conditions but did not show any influence on 5-HT-induced effects. Among mitogen-activated protein kinase (MAPK) inhibitors, a p38 MAPK inhibitor inhibited 5-HT-induced effects on colonic ICCs. Thus, 5-HT’s effect on pacemaker activity in small intestinal and colonic ICCs has excitatory but variable patterns. ANO1, T-type Ca^2+^, and HCN channels are involved in 5-HT-induced effects, and MAPKs are involved in 5-HT effects in colonic ICCs.

## 1. Introduction

Although 5-hydroxytryptamine (5-HT) is well known for its important functions in diverse organs, a high amount of 5-HT is produced in the gastrointestinal (GI) tract. In the GI system, 5-HT is involved in many functions such as enteric nervous system development, inflammation, motility, and others [1].

Many studies have shown that 5-HT contributes to excitatory synaptic transmission and this leads to propagating contractions, indicating that 5-HT in the GI system is a critical mediator of motility [2,3,4,5]. The release of 5-HT has been shown to activate the ascending and descending circuits that comprise the peristaltic reflex and cause the contraction and relaxation of smooth muscles, respectively [6,7,8]. Many experiments have revealed that 5-HT release is a key factor for normal motility. GI transit is significantly slower in terms of total, colonic, and small intestinal transit times by decreasing 5-HT biosynthesis from the nervous system, enterochromaffin cells, and mast cells in the GI tract [9,10,11,12].

The GI tract shows spontaneous contraction that is evoked by periodic membrane depolarizations referred to as slow waves. Interstitial cells of Cajal (ICCs) are pacemaker cells that generate electrical slow waves and lead to phasic contractions in the smooth muscle layers of the intestine [13]. ICCs are responsible for generating and propagating electrical slow wave activity and underlying smooth muscle contractions and mediating nitrergic and cholinergic neuromuscular neurotransmission [14]. This indicates that ICCs can be the target for neurotransmitters to regulate GI motility and understanding the role of 5-HT on colonic ICCs, the activation of which can regulate GI motility, is essential. To date, there is no report on how 5-HT acts on electrical slow waves generated by ICCs in the colon.

The contractile activities vary from the stomach to the colon in the form of different slow waves. In our previous work, we stated that this is probably attributed to the fact that the pattern of pacemaker activity in ICCs is different between the small intestine and colon [15,16]. In particular, many functional channels (ANO1, T-type Ca^2+^, and HCN channels) are involved in generating pacemaker activity in colonic ICCs because of their electrical importance in ICCs. However, to our knowledge, no study has reported the role of 5-HT on pacemaker activity and channels in colonic ICCs. Furthermore, endogenous tyrosine kinases were shown to participate in the generation of pacemaker potentials in colonic ICCs in a previous study [17], indicating the possibility that tyrosine kinases may have a role in 5-HT-induced effects on pacemaker activity in colonic ICCs. Therefore, we examined the action and mechanism of 5-HT on pacemaker activity in colonic ICCs by comparing 5-HT’s actions between the small intestinal and colonic ICCs.

## 2. Results

### 2.1. Effects of 5-HT on Pacemaker Potentials in Colonic ICCs

In the present study, we examined the functional effects of 5-HT on pacemaker potential by using cultured colonic ICCs. Cultured ICCs showed fusiform cell bodies and multiple processes of neighboring cells (Appendix A). We performed electrophysiological recording from cultured ICCs under the current mode (*I* = 0). Under controlled conditions, the resting membrane potential and pacemaker potential frequency were −58.7 ± 0.4 mV and 10.7 ± 0.5 cycles/5 min, respectively (*n* = 15). The frequency slightly increased in cells treated with 0.1 μM of 5-HT. The resting membrane potential and frequency at 0.1 μM of 5-HT were −60.4 ± 0.3 mV and 11.7 ± 0.5 cycles/5 min, respectively (*n* = 7, Figure 1a). Exposure to 1 μM of 5-HT increased the frequency and slightly induced depolarization. The resting membrane potential and frequency at 1 μM of 5-HT were −52.4 ± 0.7 mV and 20.7 ± 1.5 cycles/5 min, respectively (*n* = 7, Figure 1b). Treatment with a high dose of 5-HT (10 μM) increased the frequency of pacemaker potentials and induced depolarization of membrane potentials in colonic ICCs (*n* = 7, Figure 1c). The summarized resting membrane potentials and frequency with various doses of 5-HT are presented in Figure 1d,e. These results suggest that 5-HT has an excitatory action on the pacemaker activity of colonic ICCs.

### 2.2. Involvement of Hyperpolarization-Activated Cyclic Nucleotide-Gated Channel for 5-HT Action in Colonic ICCs

There are some reports that the HCN channel plays a role in generating pacemaker activity in colonic ICCs. To determine whether the HCN channel is involved in this 5-HT-induced effect, HCN channel inhibitors were used with 5-HT. The exposure of HCN channel blockers (10 μM of ZD7288 or 10 μM of CsCl) suppressed the spontaneous pacemaker potentials of colonic ICCs, and both blocked the 5-HT-induced effect on pacemaker activity (*n* = 5, Figure 2a,b). Owing to the importance of the intracellular cAMP for the HCN channel, we used dideoxyadenosine (DDA, an adenylate cyclase inhibitor) and cell-permeable 8-bromo-cAMP. The use of DDA (100 μM) blocked the generation of pacemaker activity in colonic ICCs, and this suppressed 5-HT-induced effects on pacemaker potentials (*n* = 5, Figure 2c). To confirm the role of cAMP, colonic ICCs were exposed to 8-bromo-cAMP (100 μM), and this resulted in depolarization of membrane potentials and increasing frequency (*n* = 5, Figure 2d). The effects of HCN channel blockers and DDA or 8-bromo-cAMP on 5-HT-induced effects are summarized in Figure 2e,f. These results suggest that the HCN channel is important for pacemaker activity in colonic ICCs under normal conditions and is needed for 5-HT effects.

### 2.3. Involvement of Various 5-HT Receptors on 5-HT Action in Colonic ICCs

To understand which receptor is involved in 5-HT action in colonic ICCs, we tested specific agonists in colonic ICCs. Each agonist was tested with various concentrations from 0.1 to 20 10 μM and tested with the proper concentration. To determine the functional role of the 5-HT_2B_ receptor, we evaluated the effect of a selective 5-HT_2B_ receptor agonist (BW-723C86, 10 μM). Exposure to BW-723C86 increased the frequency of pacemaker potentials and depolarized the membrane potentials of colonic ICCs (*n* = 6, Figure 3a). Next, we used cisapride (10 μM) or prucalopride (30 μM), which are 5-HT_4_ receptor agonists. Both increased the frequency of pacemaker activity, and cisapride showed strong depolarization of membrane potentials in colonic ICCs (*n* = 6, Figure 3b,c). Exposure to LP-211 (10 μM), which is a 5-HT_7_ agonist in colonic ICCs, produced an increased frequency of pacemaker activity (*n* = 6, Figure 3d). The effects of 5-HT receptor agonists on pacemaker potentials in colonic ICCs are presented in Figure 3e,f.

### 2.4. Effects of 5-HT on Pacemaker Potentials in Small Intestinal ICCs

To carry out a comparative study between small intestinal and colonic ICCs, we tested the effect of 5-HT on small intestinal ICCs. We performed electrophysiological recording of small intestinal ICCs under the current mode (*I* = 0). Under controlled conditions, the resting membrane potential and pacemaker potential frequency were −64.6 ± 0.2 mV and 14.4 ± 0.3 cycles/min, respectively (*n* = 11). Treatment with 5-HT (10 μM) resulted in transient depolarization of the membrane potentials. Moreover, during transient depolarization, the pacemaker potential frequencies did not change or increased slightly (*n* = 11, Figure 4a). To understand the role of cAMP on small intestinal ICCs, we used DDA (100 μM). DDA did not show any influence on the pacemaker activity of small intestinal ICCs, and the effect of 5-HT on small intestinal ICCs persisted (*n* = 5, Figure 4b). The effects of these substances on 5-HT-induced actions are presented in Figure 4c,d.

### 2.5. No Involvement of Hyperpolarization-Activated Cyclic Nucleotide-Gated Channel in 5-HT-Mediated Actions in Small Intestinal ICCs

To determine whether the HCN channel is involved in the 5-HT-induced effect on small intestinal ICCs, we tested HCN blockers with 5-HT. Exposure to HCN channel blockers (10 μM of ZD7288, 10 μM of Zatebradine, or 10 μM of CsCl) did not produce any effect on the pacemaker potentials of the small intestinal ICCs. Furthermore, 5-HT (10 μM) showed transient depolarization in the presence of HCN blockers (*n* = 5, Figure 5a–c). The effects of HCN channel blockers on 5-HT-induced effects are presented in Figure 5d,e.

### 2.6. Involvement of ANO1 and T-Type Ca^2+^ Channels for 5-HT Action in Colonic ICCs

To determine whether ANO1 and T-type Ca^2+^ channels are involved in 5-HT-induced effects, the inhibitor of ANO1 or T-type Ca^2+^ channels was used with 5-HT. Treatment with ANO1 channel blockers (5 μM of CaCCinh-A01 or 5 μM of T16Ainh-A01) inhibited the generation of pacemaker potentials in colonic ICCs and blocked the 5-HT-induced augmentation of pacemaker activity (*n* = 6–7, Figure 6a,b). Moreover, ML218 (T-type Ca^2+^ channel blocker, 5 μM) also inhibited the generation of pacemaker potentials and blocked 5-HT-induced actions on pacemaker activity (*n* = 7, Figure 6c). The effects of the ANO1 or T-type Ca^2+^ channel blocker on 5-HT-induced effects are summarized in Figure 6d,e. These results may imply that the activation of ANO1 and T-type Ca^2+^ channels is essential for the generation of pacemaker activity and that they are important for 5-HT’s effects on pacemaker activity in colonic ICCs.

### 2.7. Effects of Tyrosine Kinases or Mitogen-Activated Protein Kinases on Colonic ICCs

To determine the role of tyrosine kinases or mitogen-activated protein kinases (MAPKs) on normal pacemaker activity and 5-HT-induced effects on colonic ICCs, tyrosine kinase and MAPK inhibitors were used with 5-HT. Exposure to genistein (10 μM, a tyrosine kinase inhibitor) inhibited the pacemaker activity of colonic ICCs but did not show any influence on 5-HT-induced effects (*n* = 6, Figure 7a). Treatment with PD 98059 (10 μM, a selective p42/44 inhibitor) did not show any influence on pacemaker activity and 5-HT-induced effects on pacemaker activity in colonic ICCs (*n* = 7, Figure 7b); however, SB 203580 (10 μM, a p38 MAPK inhibitor) itself decreased normal pacemaker activity and blocked the 5-HT-induced effects (*n* = 7, Figure 7c). The exposure of c-jun NH_2_-terminal kinase (10 μM, JNK)-II inhibitor inhibited the pacemaker activity but did not have an influence on 5-HT-induced effects on colonic ICCs (*n* = 7, Figure 7d). The effects of tyrosine kinase inhibitors or MAPK blockers on 5-HT-induced effects are presented in Figure 7e,f.

## 3. Discussion

Significant quantities of 5-HT are found in enterochromaffin cells, serotonergic neurons, platelets, and mast cells. There are 5-HT transporters in serotonergic neurons, mucosal cells, and submucosal cells, all of which uptake 5-HT. As a result, ICCs and myenteric neurons respond to 5-HT released from mucosa [18]. Thus, 5-HT can act on ICCs and myenteric neurons in the muscle layer. In this study, we found that 5-HT produced the depolarization of membrane potentials and increased the frequency of pacemaker activity in colonic ICCs. The way that 5-HT affects colonic ICCs differs from the way that it affects small intestine ICCs. In Figure 4, the effects of 5-HT on pacemaker potentials in small intestinal ICCs showed depolarization of membrane potentials; however, this occurred transiently and the frequency increased slightly. In colonic ICCs, 5-HT caused tonic depolarization with strongly increasing frequencies. Considering these findings, we hypothesized that the differing effect pattern of 5-HT on small and colonic ICCs may be attributed to different mechanisms. The main concern of this study is that 5-HT’s action on pacemaker activity is caused by acting on small intestinal or colonic ICCs directly or not because there are other types of cells like neurons, smooth muscle cells, etc., in cultured cells. However, ANO1 is a specific channel in ICCs and we found that ANO1 blockers inhibited the pacemaker activity of ICCs in this study. This may be evidence that the target of 5-HT for acting pacemaker activity is ICCs.

In our previous study, we showed that the HCN channel is present in colonic ICCs, and this regulates the pacemaker activity in colonic ICCs. In addition, we showed that the HCN channel does not play a role in small intestinal ICCs [19]. To verify these findings, we tested and found that HCN blockers inhibited the pacemaker activity in colonic ICCs and blocked the 5-HT-induced effects. However, they did not inhibit normal pacemaker activity and did not show any influence on 5-HT-induced effects in small intestinal ICCs. Moreover, our previous study showed that basal cAMP production was higher in colonic than in small intestinal ICCs [19]. In this work, we also showed that the adenylate cyclase inhibitor or cell-permeable 8-bromo-cAMP showed effects on colonic ICCs but not on small intestinal ICCs, indicating that cAMP in colonic ICCs is an important messenger for normal pacemaker activity and can be the target for exogenous substances. Intracellular cAMP is important for opening the HCN channel [20]. Adenylate cyclase activity and phosphodiesterase determine intracellular cAMP concentration [21]. These facts strongly suggest the possibility that cAMP is a crucial messenger for 5-HT-induced effects in colonic ICCs, and this could explain why the HCN channel is important for regulating pacemaker activity in colonic ICCs by maintaining a high amount of cAMP. In our previous study, we suggested that adenylate cyclase type 1, which is a Ca^2+^-dependent adenylate cyclase, was detected in colonic ICCs [15]. Our findings support this suggestion.

Interestingly, 5-HT receptors can be divided into seven forms (from 5-HT_1_ to 5-HT_7_), which activate an intracellular second messenger to produce an excitatory or inhibitory response [22]. 5-HT_2_, 5-HT_3_, 5-HT_4_, and 5-HT_7_ have excitatory effects in diverse tissues [20], and the stimulation of those receptors may lead to the production of the intracellular signaling molecule cAMP [23]. In addition, it has been reported that 5-HT_1_, 5-HT_2_, 5-HT_3_, 5-HT_4_, and 5-HT_7_ are present in the intestine [24]. In this study, to examine why 5-HT showed excitatory functional action on pacemaker activity in colonic ICCs, we measured the effects of specific agonists for various 5-HT receptors. Our results showed that all agonists for 5-HT_2B, 4, and 7_ receptors, such as 5-HT, had functional effects on pacemaker activity in colonic ICCs. This indicates that many 5-HT receptors respond to 5-HT in colonic ICCs and, among them, we were interested in examining the 5-HT_4_ receptor because it is a G protein-coupled receptor that stimulates cAMP production in response to 5-HT [25]. In our previous study, we strongly suggested the importance of cAMP in colonic ICCs [15]. Cisapride is a gastroprokinetic agent that increases motility by activating the 5-HT_4_ receptor [26]. In this study, we found that cisapride showed dramatic depolarization and increased the frequency of pacemaker activity in colonic ICCs. To confirm this, we also tested prucalopride, which has a higher affinity for the 5-HT_4_ receptor, and found excitatory action in colonic ICCs. In this study, we also found that cisapride-induced effects on pacemaker activity are more potent than prucalopride. We think this is because cisapride can stimulate ACh from the enteric nervous system. To date, it has been reported that 5-HT_1, 2, 3, 4, and 7_ receptors are expressed in the GI tract and affect GI motor function [18]. In our previous study, we showed that 5-HT modulated pacemaker activity in small intestinal ICCs through 5-HT_3_, 5-HT_4_, and 5-HT_7_ receptors but not through 5-HT_2_ [16]. Moreover, 5-HT has excitatory effects in small intestinal and colonic ICCs, but the different effect of the 5-HT receptor provides the possibility that the acting mechanism is different in small intestinal and colonic ICCs.

Many reports have suggested ANO1, T-type Ca^2+^, HCN channels, or Na^+^/Ca^2+^ exchangers as representative ion channels or transporters for the generation of pacemaker activity in ICCs [27,28]. To date, although the exact mechanisms for the generation of pacemaker activity remain unknown, it has been reported in many studies that the aforementioned are involved in pacemakers [29,30,31]. Among these mechanisms, we determined the importance of those channels on 5-HT-induced effects in colonic ICCs. In the presence of ANO1 or T-type Ca^2+^ channel inhibitors, the pacemaker activity stopped in colonic ICCs, and, then, these blocked 5-HT-induced effects. This indicates that ANO1, T-type Ca^2+^, and HCN channels are essential for 5-HT effects on pacemaker activity. Furthermore, for the generation of pacemaker activity, many channels or transporters are needed. To understand how ANO1 or T-type Ca^2+^ is involved in 5-HT-induced effects, further studies are needed.

Receptor tyrosine kinases are important mediators of the signaling transduction process, leading to diverse biological processes in response to external and internal stimuli [32]. Several studies considering a potential role for tyrosine kinases as regulators of HCN channels have reported that channel function is altered by tyrosine kinase inhibitors [33,34,35]. Furthermore, we showed that endogenous tyrosine kinases participate in the generation of pacemaker potentials in colonic, but not in small intestinal, ICCs [17]. This indicates the possibility that tyrosine kinases may have a role in 5-HT-induced effects on pacemaker activity in colonic ICCs. Moreover, we found that a tyrosine kinase inhibitor blocked the normal pacemaker activity but did not show any action on 5-HT-induced effects (Figure 7). This means that tyrosine kinases regulate the pacemaker activity under controlled conditions, but they do not have an impact on the 5-HT-induced effects in colonic ICCs. Next, we determined the role of MAPKs in colonic ICCs. MAPKs are important intracellular signaling substances because of their significant roles in diverse cells, including smooth muscle contraction [36]. There are reports on the modulation of the HCN channel by MAPKs, especially by p38 MAPK, on the cellular level. The HCN channel may be modulated by p38 MAPK in the hippocampal and neocortical pyramidal neurons, among others [37], and the HCN channel may be gated via MAPK activation by protein kinase C [38]. Furthermore, we found that MAPK inhibitors inhibited the 5-HT-induced effects in small intestinal ICCs, indicating that MAPKs may be involved in the modulation of pacemaker activity by 5-HT in our previous study [16]. To understand this in colonic ICCs, we tested various kinds of MAPK inhibitors and found that p38 MAPK or JNK-II inhibitors inhibited the pacemaker activity under controlled conditions in colonic ICCs. In addition, only a p38 MAPK was involved in 5-HT-induced effects but not JNK-II. This suggests that MAPKs also can be modulators for pacemaker activity and 5-HT function in colonic ICCs. Further research is needed to determine the precise mechanism of modulation or the substance involved in MAPKs and the HCN channels.

## 4. Materials and Methods

### 4.1. Ethical Approval and Experimental Animals

The experiments were performed according to the Guiding Principles of the National Institutes of Health Guide for the Care and Use of Laboratory Animals and were approved by the Ethics Committee of Chosun University. Mice were fed a standard diet and had free access to water. ICR mice (5–8 days old) of either sex were anesthetized or sacrificed using isoflurane. Every effort was made to minimize the number of animals used and their suffering. In total, 85 ICC mice were used for the study and the number of each data is the same as the number of mice.

### 4.2. Preparation of Cells 

The colon and small intestine were removed from below the duodenum to the rectum. The whole portion of the small intestine or colon was used. The colon and small intestine were opened along the mesenteric border. Opened tissues were pinned to the bottom of a Sylgard dish filled with ice-cold Ca^2+^-free Hank’s solution. After removing the luminal contents, the mucosa and submucosa were removed by dissection. The isolated tissue was equilibrated in Ca^2+^-free Hank’s solution for 30 min. Cells were scattered with an enzyme solution containing 1.3–1.8 mg/mL collagenase (Worthington Biochemical Co., Lakewood, NJ, USA), 2 mg/mL bovine serum albumin (Sigma, St. Louis, MO, USA), 2 mg/mL trypsin inhibitor (Sigma), and 0.27 mg/mL adenosine triphosphate and then incubated in a water bath at 37 °C for 12–14 min. After three washes with Ca^2+^-free Hank’s solution to remove enzymes, the tissues were triturated by a series of blunt pipettes of decreasing tip diameter. Small pieces of tissue were placed onto sterile glass coverslips coated with 200 μL poly-L-lysine (Sigma-Aldrich, St. Louis, MO, USA) in 35 mm culture dishes. Cells were cultured in smooth muscle growth medium (Medium 231; Gibco, Grand Island, NY, USA) supplemented with 1% antibiotics/antimycotics (Gibco) and 5 ng/mL urine stem cell factor (SCF; Sigma) at 37 °C/5% CO_2_. 

### 4.3. Solutions and Drugs

The cells were bathed in a solution containing 5 mM KCl, 135 mM NaCl, 2 mM CaCl_2_, 10 mM glucose, 1.2 mM MgCl_2_, and 10 mM HEPES adjusted to pH 7.2 with Tris. The pipette solution contained 140 mM KCl, 5 mM MgCl_2_, 2.7 mM K_2_ATP, 0.1 mM Na_2_GTP, 2.5 mM creatine phosphate disodium, 5 mM HEPES, and 0.1 mM EGTA; the solution was adjusted to pH 7.2 with Tris.

All drugs were purchased from Sigma Chemical Co. (St. Louis, MO, USA) (Cat no. 5-HT; H9523, ZD7288; Z3777, CsCl; C4036, DDA; 288104, 8-bromo-cAMP; B5386, BW723C86; B175, cisapride; 1134120, prucalopride; SML1371, LP-211; SML1561, zatebradine; Z0127, CaCCinh-A01; SML0916, T16Ainh-A01; 613551, ML218; SML0385, genistein; G6649, PD98059; P215, SB203580; S8307, JNK inhibitor II; 420128) and dissolved in distilled water or dimethyl sulfoxide (DMSO) to prepare stock solutions (10 or 100 mM), which were either added to the bath solution or applied to whole-cell preparations using superfusion. The final DMSO concentration was less than 0.05%. The concentration of agonists or antagonists was determined according to previous reports [15,16,18,27]. 

### 4.4. Patch Clamp Experiments

The current clamp mode was used to record the pacemaker potentials in the stomach, small intestine, and colon ICCs when a network-like structure appeared after culturing (1–3 days). The pacemaker potentials were amplified using an Axopatch 200 B (Axon Instruments, Foster, CA, USA). Data were filtered at 5 kHz and displayed on a computer. The results were analyzed using pClamp and GraphPad Prism (version 5.0; GraphPad Software Inc., San Diego, CA, USA). All operations were performed at 30 °C.

### 4.5. Statistical Analysis

Data are expressed as the mean ± standard error of the mean (SEM). Differences in the data were evaluated using Student’s *t*-test and analysis of variance (ANOVA) followed by a post hoc test. Differences with *p*-values less than 0.05 were considered statistically significant. The *n* values reported in the text refer to the number of cells used in the patch-clamp experiments.

## 5. Conclusions

In conclusion, many studies have revealed the important excitatory roles of 5-HT in GI motility, but the mechanisms underlying this response have proven to be increasingly complex, especially in ICCs. In this study, we showed that the effects of 5-HT on pacemaker activity in small intestinal and colonic ICCs are excitatory, but have a different pattern. ANO1, T-type Ca^2+^, and HCN channels are involved in 5-HT-induced effects, and MAPKs are partially associated with 5-HT-induced effects in colonic ICCs.

## Figures and Tables

**Figure 1 ijms-25-03997-f001:**
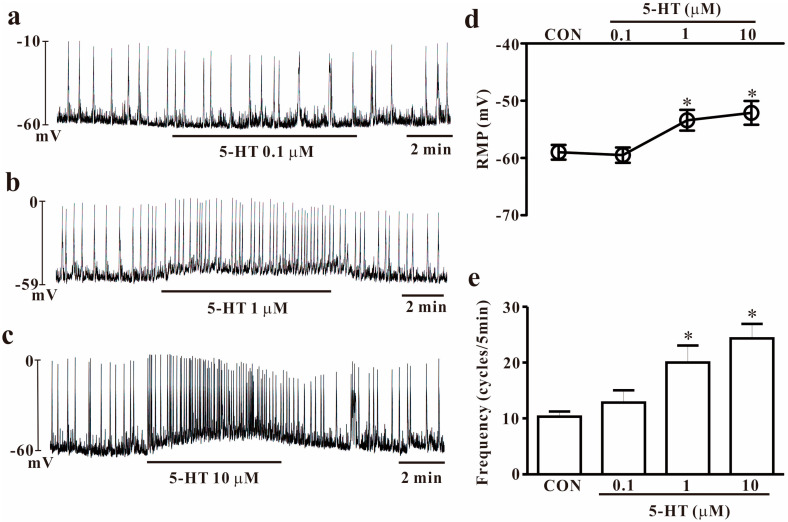
Effects of 5-HT on pacemaker potentials of cultured colonic ICCs in mice. (**a**–**c**) The pacemaker potentials of ICC exposed to 5-HT (1–10 μM) in the current clamp mode. 5-HT depolarizes the membrane and increases the pacemaker potential frequency in a dose-dependent manner. (**d**,**e**) Summary of excitatory effects of 5-HT on pacemaker potentials. Each column represents the mean ± SEM. Asterisks indicate values that are significantly different from the control values (*p* < 0.05). CON: control; RMP: resting membrane potential. RMP or frequency indicates the maximum membrane potentials or average frequency for 5 min during treatment.

**Figure 2 ijms-25-03997-f002:**
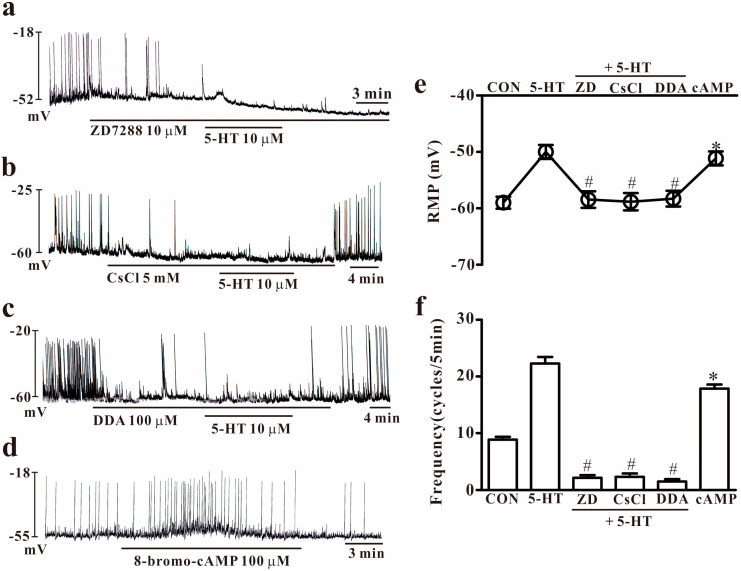
Effects of hyperpolarization-activated cyclic nucleotide-gated (HCN) blockers and adenylate cyclase inhibitors on 5-HT-induced actions and effects of 8-bromo-cAMP on pacemaker potentials in colonic ICCs. (**a**,**b**) ZD7288 (10 μM) and CsCl (5 mM), both HCN channel inhibitors, inhibited the pacemaker potential and blocked 5-HT (10 μM)-induced effects in colonic ICCs. (**c**) Dideoxyadenosine (100 μM), an adenylate cyclase inhibitor, inhibited the pacemaker potential and blocked 5-HT (10 μM)-induced effects in colonic ICCs. (**d**) Cell permeable 8-bromo-cAMP (100 μM) increased the frequency of pacemaker potentials in colonic ICCs. (**e**,**f**) Summary of the effects of 5-HT on pacemaker potential and frequency in colonic ICCs. Each column represents the mean ± SEM. Asterisks or hashtags indicate values that are significantly different from the values of control (*) or 5-HT alone (#) (*p* < 0.05). CON: Control; ZD: ZD7288; DDA: dideoxyadenosine; cAMP: 8-bromo-cyclic adenosine monophosphate; RMP: resting membrane potential. RMP or frequency indicates the maximum membrane potentials or average frequency for 5 min during treatment.

**Figure 3 ijms-25-03997-f003:**
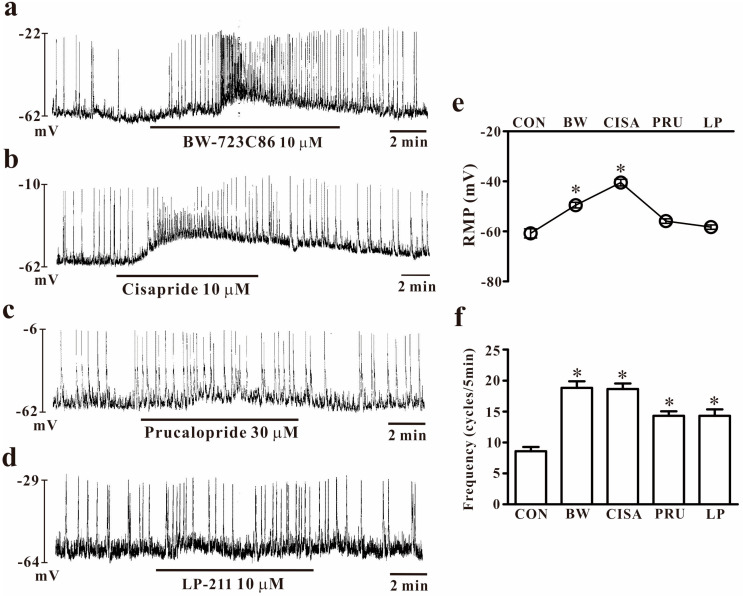
Effects of 5-HT receptor agonists on pacemaker potential in colonic ICCs. (**a**) BW-723C86 (10 μM), a 5-HT_2B_ receptor agonist, depolarized the membrane potentials and increased the pacemaker potential frequency. (**b**,**c**) Cisapride (10 μM) and prucalopride (30 μM), which are specific agonists of 5-HT_4_ receptors, depolarized the membrane potentials and increased the pacemaker potential frequency. (**d**) LP-211 (10 μM), a specific agonist of the 5-HT_7_ receptor, resulted in slight depolarization of the membrane potentials and increased the frequency of pacemaker activity. (**e**,**f**) Summary of 5-HT receptor agonists on pacemaker activity in colonic ICCs. Each column represents the mean ± SEM. Asterisks indicate values that are significantly different from the control values (*p* < 0.05). CON: control; BW: BW-723C86; CISA: cisapride; PRU: prucalopride; LP: LP-211; RMP: resting membrane potential. RMP or frequency indicates the maximum membrane potentials or average frequency for 5 min during treatment.

**Figure 4 ijms-25-03997-f004:**
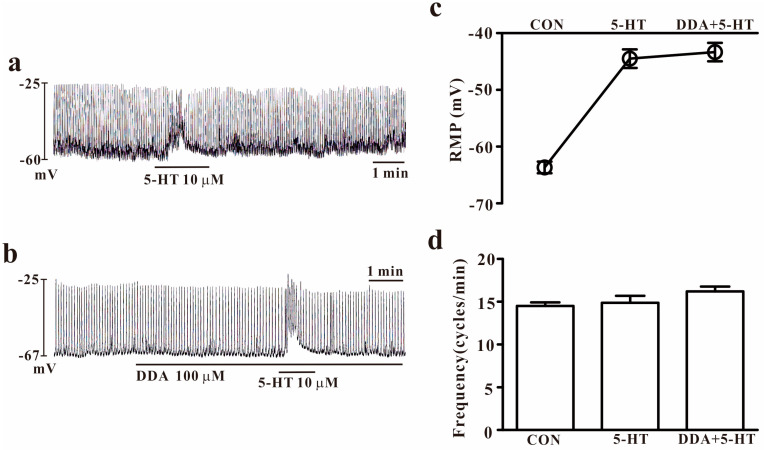
Effects of 5-HT on pacemaker potential and adenylate cyclase inhibitors on 5-HT-induced effects in small intestinal ICCs. (**a**) 5-HT (10 μM) generated transient depolarization of membrane potentials in small intestinal ICCs. (**b**) Dideoxyadenosine (100 μM) is an adenylate cyclase inhibitor and did not affect the pacemaker potential or block 5-HT (10 μM)-induced effects in small intestinal ICCs. (**c**,**d**) Summary of the effects of each substrate, with or without 5-HT, on the pacemaker activity in small intestinal ICCs. Each column represents the mean ± SEM. CON: control; DDA: dideoxyadenosine; RMP: resting membrane potential. RMP or frequency indicates the maximum membrane potentials or average frequency for 5 min during treatment.

**Figure 5 ijms-25-03997-f005:**
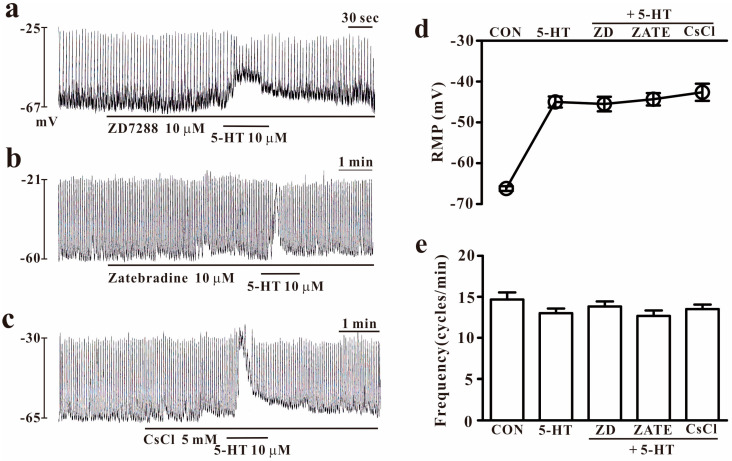
Effects of HCN blockers on 5-HT-induced changes in small intestinal ICC cells (**a**–**c**). ZD7288 (10 μM), zatebradine (10 μM), and CsCl (5 mM) are HCN channel inhibitors and did not demonstrate any effects on the pacemaker potential nor did they block 5-HT (10 μM)-induced effects in small intestinal ICCs. (**d**,**e**) Summary of the effects of HCN blockers and 5-HT on the pacemaker activity in small intestinal ICCs. Each column represents the mean ± SEM. CON: control; ZD: ZD7288; ZATE: zatebradine; HCN: hyperpolarization-activated cyclic nucleotide-gated; RMP: resting membrane potential. RMP or frequency indicates the maximum membrane potentials or average frequency for 5 min during treatment.

**Figure 6 ijms-25-03997-f006:**
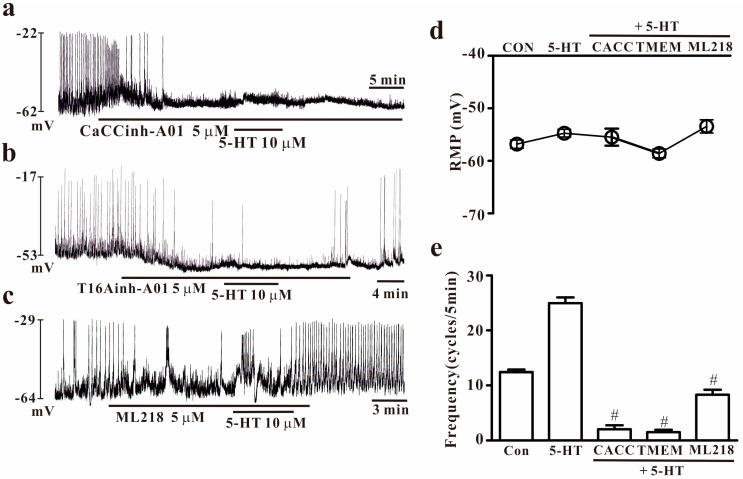
Effects of ANO1 and T-type Ca^2+^ channel inhibitors on 5-HT’s actions in colonic ICCs. (**a**,**b**) Ca^2+^-activated Cl^−^ channel (5 μM of CaCCinh-A01) or TMEM16A channel (5 μM of T16Ainh-A01) inhibitor blocked the pacemaker activity and inhibited 5-HT (10 μM)-induced effects on pacemaker potentials in colonic ICCs. (**c**) ML218 (5 μM), a T-type Ca^2+^ channel inhibitor, inhibited the pacemaker activity and 5-HT-induced effects in colonic ICCs. (**d**,**e**) Summary of 5-HT receptor agonists on pacemaker potential frequencies in colonic ICCs. Each column represents the mean ± SEM. Hashtags indicate values that are significantly different from the values of 5-HT alone (*p* < 0.05). CON: control; CACC: CaCCinh-A01; TMEM: T16Ainh-A01; RMP: resting membrane potential. RMP or frequency indicates the maximum membrane potentials or average frequency for 5 min during treatment.

**Figure 7 ijms-25-03997-f007:**
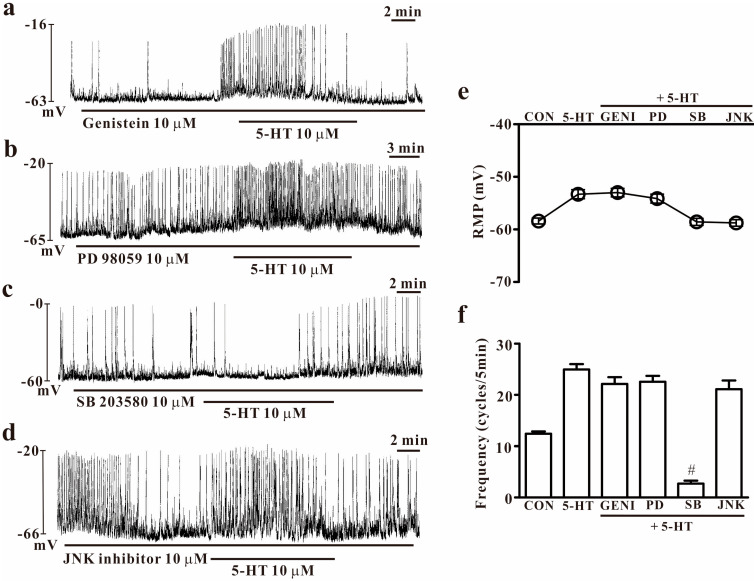
Effects of tyrosine kinase and MAPK inhibitors on 5-HT-induced effects in colonic ICCs. (**a**) Genistein (10 μM), a tyrosine kinases inhibitor, blocked the pacemaker activity of colonic ICCs but did not affect 5-HT-induced effects on pacemaker activity. (**b**) PD 98059 (10 μM), a selective p42/44 inhibitor, did not affect 5-HT-induced effects on pacemaker activity in colonic ICCs. (**c**) SB 203580 (10 μM), a p38 MAPK inhibitor, inhibited the pacemaker activity under control conditions in colonic ICCs and inhibited 5-HT (10 μM)-induced effects. (**d**) c-jun NH_2_-terminal kinase (JNK)-II inhibitor (10 μM) inhibited the pacemaker activity under control conditions in colonic ICCs but did not influence 5-HT (10 μM)-induced effects. (**e**,**f**) Summary of the effects of 5-HT on pacemaker potential frequency in colonic ICCs. Each column represents the mean ± SEM. Hashtags indicate values that are significantly different from the values of 5-HT alone (*p* < 0.05). CON: control; GENI: genistein; MAPK, mitogen-activated protein kinase; PD: PD 98059; SB: SB 203580; JNK: JNK-II inhibitor; RMP: resting membrane potential. RMP or frequency indicates the maximum membrane potentials or average frequency for 5 min during treatment.

## Data Availability

The data presented in this study are available upon request from the corresponding author.

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
