# Peer review of "5-Hydroxytryptamine Enhances the Pacemaker Activity of Interstitial Cells of Cajal in Mouse Colon"

_ijms, 2024, doi:10.3390/ijms25073997_

Round 1

Reviewer 1 Report (New Reviewer)

Comments and Suggestions for Authors

The authors used a lot of pharmacological tools to study the enhancive effect of 5-HT on colonic ICC pacemaker activity and the underlying mechanisms. Patch-clamp under the current mode (I=0) was used to record the membrane potential oscillation from in vitro cultured ICCs. However, the current version still needs a major revision to reach a good quality to publish. 

Comments:

1.     The MS was not well prepared, the writing needs to improve.  The introduction part is too brief to help readers have a good understanding of the motivation of this study.

2.     In vitro culture could change the properties of ICCs, especially after long-term culture. How long had the cells been cultured before patch-clamp recording? In addition, there are methods to freshly isolate single cells from the gastrointestinal smooth muscle layer for patch-clamp recording (for example, PMID: 27101932, PMID: 19967074). It’s better to do at least one experiment with freshly isolated ICCs to validate if cultured ICCs showed similar properties. 

3.     After culture, how did you distinguish ICCs from other cell types (such as the abundant smooth muscle cells)? If only based on the morphology of the cells, representative photos of cultured ICCs (immunofluorescent with c-Kit antibody will be a great validation) and other cell types should be provided. 

4.     The cells were cultured from small pieces of tissue. Based on my personal experience, primary cultured cells outgrowth from tissue pieces can still interact with each other. So, how could we distinguish whether the effects of 5-HT and other drugs are directly from ICCs? Or indirectly from other cell types (like smooth muscle cells) interacted with ICC?

5.     In lines 18  19, please double-check the expression of this sentence; the example should be an agonist instead of a receptor.

6.     Lines 56 – 57, the statement is incorrect. There are previously published papers that reported the role of 5-HT on pacemaker activity of colonic ICCs, for example, PMID: 33895170 and PMID: 25807879. If the authors want to emphasize the novelty of this research, please describe clearly, for example “there is no study used patch-clamp recording…”.

7.     Figure 1b, is not a good representative recording trace. Before the treatment with 1 μM 5-HT was already higher than -55 mV, and didn’t show obvious elevation during the treatment.

8.     I guess the n number in lines 66, 68, 70, and 72 represents the cell number, but how many animals were the recorded cells from?

9.     Please introduce the HCN channel function and its role in ICC pacemaker activity, either in the introduction or in result 2.2. People read papers from front to back, and the late description in the discussion makes no sense to help people understand the authors’ logic. It’s also necessary to introduce other drug targets studied in this MS.

10.  Lines 86 – 87, need to cite reference papers to support ZD7288 and CsCl, and the concentration used in this study is specific to block HCN channels. Citations are also required to support the use of other inhibitors, blockers, agonists, antagonists, and other chemicals.

11.  Result 2.3 part. Please include citations supporting agonists and concentrations in this study specific to certain 5-HT receptors.

12.  Figure 2.  Based on the legend, asterisks indicate statistical significance compared with the control. However, in panel e, ZD + 5-HT, CsCl + 5-HT, and DDA + 5-HT didn’t show an obvious change in the RMP if compared with CON. Actually, these 3 groups should be compared with the 5-HT only group.

13.  Line 142: why was SQ22536 not used in the colonic ICC study?

14.  Figure 4, based on the representative recording traces, there is a transient increase in frequency since stimulating with 5-HT, the quantification didn’t match this phenomenon. In addition, the 5-HT treatment duration is much shorter than experiments with colonic ICCs. I suggest the authors keep using the same stimulation protocol for colonic ICCs.

15.  Figure 5, the 5-HT treatment duration is also much shorter than experiments with colonic ICCs.

16.  Result 2.6. There are many kinds of Ca2+-activated Cl- channel inhibitors and TMEM16A inhibitors, please indicate the names of the ones used in this study. 

17.  Result 2.7. Please add an introduction about the function of tyrosine kinases and MAPKs in ICC pacemaker activity with suitable citations. 

18.  Prucalopride was described as a 5-HT4 receptor agonist with higher affinity. But why did it show a milder effect than cisapride, since the concentration (30 μM) used for prucalopride was even higher?

19.  The method part needs more details.

20.  Lines 318 – 319, please clearly describe the position of the small intestine and colon tissues used for ICC isolation. It’s better to describe the colon and small intestine, respectively.

21.  Please clearly describe the criteria for RMP quantification. Especially for the 5-HT and 8-bromo-cAMP treatment, did you use the average RMP or the highest RMP during the treatment?

22.  Please also describe the criteria for frequency quantification. Based on the representative traces, during some of the treatments (for example, 8-bromo-cAMP), the frequency change wasnt sustained; the quantification criteria are critical to describing the change.

23.  Please provide the Cat number for all the drugs used in this study. The solvent used for each drug and the concentration of stock solutions also need to be indicated. 

24.  Only Figure 1d and 1e are suitable for using one-way (the factor is the concentration of 5-HA) ANOVA + post hoc test. The authors also need to indicate which post hoc test was used. For quantifications in other Figures, please consult a statistician. Based my experience, should first do a normality test, if the data distributed normally, compare each experimental group with the no-treatment control or 5-HT-only group with t-test based on needs.

Comments on the Quality of English Language

1.     The writing of this MS needs to improve. The results part of this MS reads like a fragmented data set; paragraphs have no logical connections.

2.     There are also some grammar problems in the MS. For example, in line 37, ‘this leads propagating contractions’ should be ‘ this leads to propagating contractions’. In line 43, ‘the’ should be added before ‘nervous system’ and ‘GI tract’. As the authors are not native English speakers, I suggest them to use some professional English polishing service.

Author Response

Thank you for giving me the opportunity to submit a revised draft of our manuscript titled “5-Hydroxytryptamine Enhances the Pacemaker Activity of Interstitial Cells of Cajal in Mouse Colon” to International Journal of Molecular Sciences. We appreciate the time and effort that you and reviewers have dedicated to providing your valuable feedback on my manuscript. We have checked carefully with reviewers’ comments and corrected as their suggestions. For submitting our revisions, we have prepared the following documents and the important changes made in the manuscript are marked with blue color. Furthermore, we got English editing service from MDPI. We have attached the certification document.

Reviewer 2 Report (New Reviewer)

Comments and Suggestions for Authors

The manuscript “5-Hydroxytryptamine Enhances the Pacemaker Activity of Interstitial Cells of Cajal in Mouse Colon” by Huang is a research article which examined the localization of 5-hydroxytryptamine (5-HT) receptor and its effects on mouse colonic interstitial cells of Cajal (ICCs) using electrophysiological methods. The authors found that 5-HT increased the pacemaker activity in colonic ICCs, which was blocked by hyperpolarization-activated cyclic nucleotide-gated (HCN) channel blockers. In addition, the authors found that an adenylate cyclase inhibitor inhibited 5-HT-induced effects and cell-permeable 8-bromo-cAMP increased the pacemaker activity. Moreover, the authors demonstrated that Anoctamin-1 (ANO1), T-type Ca2+, and HCN channels are involved in 5-HT-induced effects, and MAPKs are involved in 5-HT effects in colonic ICCs. In general, this review article is critical in this field and contains essential contents. However, I have several comments before this manuscript is accepted for publication.

1. In bar and plot graphs, all the data plots should be displayed if possible. The readers can obtain more information from these data.

2. The authors used ANOVA for multiple comparison. Please add F values in the text or figure legends.

3. Figures 2-7. All the data points were connected by lines. If these data were not obtained from one cell, the data points should not be connected by lines.

4. Figure 2. The application of ZD7288, CsCl or DDA alone reduced the frequency. The authors should add the bar graphs which show the effects of ZD7288, CsCl or DDA alone. However, in Figure 5, the application of ZD7288 or CsCl alone had almost no effects on the frequency. Please explain the discrepancy.

5. Figure 6. The application of CACC, TMEM or ML218 alone reduced the frequency. The authors should add the bar graphs which show the effects of CACC, TMEM or ML218 alone.

Author Response

Thank you for giving me the opportunity to submit a revised draft of our manuscript titled “5-Hydroxytryptamine Enhances the Pacemaker Activity of Interstitial Cells of Cajal in Mouse Colon” to International Journal of Molecular Sciences. We appreciate the time and effort that you and reviewers have dedicated to providing your valuable feedback on my manuscript. We have checked carefully with reviewers’ comments and corrected as their suggestions. For submitting our revisions, we have prepared the following documents and the important changes made in the manuscript are marked with blue color. Furthermore, we got English editing service from MDPI. We have attached the certification document.

Round 2

Reviewer 1 Report (New Reviewer)

Comments and Suggestions for Authors

I thank the authors for the new version of MS and their response to my comments. However, some of the point-by-point responses do not address my concerns, and some of them are incorrect. Please find my comments (green color) on these responses in the attachment. 

Comments on the Quality of English Language

The quality of the language is fine.

Author Response

We greatly appreciate the reviewers’ and editor’s careful reading of our revised manuscript and thoughtful comments. We believe the 2nd revision will help us improve it to a better scientific level. We have checked carefully with reviewers’ comments and corrected as their suggestions. For submitting our revisions, we have prepared the following documents and the changes made in the manuscript are marked with red shadow.

Reviewer 2 Report (New Reviewer)

Comments and Suggestions for Authors

The authors addressed my concerns.

Author Response

We appreciate the time and effort that you have dedicated to providing your valuable feedback on my manuscript. We have checked carefully with reviewer 1’s comments and corrected as his suggestions.

Round 3

Reviewer 1 Report (New Reviewer)

Comments and Suggestions for Authors

1.As you mentioned, “When we started the work with cultured ICCs from colon, we really checked the morphology using c-Kit antibody and found the shape is same with small intestinal ICCs in 2015.”, I don’t think you need more than 10 days to add a photo simply. You also mentioned here that “Whenever we submit our MS in many Journal, most of reviewers asked about this.”, that means it’s important. It would be helpful if you could even provide a phase contrast microscope photo to show the morphology of the cells you record. I just suggest a minor revision this time, but I still want the authors to add a photo of the cells they recorded.

2. In the revised Fig. 6 legend (line 204), the authors wrote, “Asterisks indicate values that are significantly different…”, however, they used “#”, the hashtag. 

Author Response

Dear Editor and Reviewer #1

We appreciate the reviewers’ and editor’s careful reading of our revised manuscript and thoughtful comments. We believe the 3rd revision will help us improve it. We have checked carefully with reviewers’ comments and corrected as their suggestions. For submitting our revisions, we have prepared the following documents and the changes made in the manuscript are marked with red shadow.

This manuscript is a resubmission of an earlier submission. The following is a list of the peer review reports and author responses from that submission.

Round 1

Reviewer 1 Report

Comments and Suggestions for Authors

This is a very disappointing manuscript.  The authors do nice jobs of presenting their electrophysiological data and their pharmacological analysis.  On the other hand, they do a poor job of discussing the physiology of the bowel and the roles that 5-HT plays in that process.  Their coverage of those subjects suggests an actual lack of understanding of either.  The problems are apparent early and are illustrated in the second paragraph of the introduction.  Part of the difficulty here may be that the authors are not native speakers of English; consequently, some sentences, in this paragraph and many others scattered throughout the text, make no sense.  For example, the authors write: “GI transit is significantly slower in terms of the total transit time, and colonic and small intestinal transit [9, 10].”  The authors forgot in this sentence to specify what is slowing GI transit.  Citation #9 is relevant to the speed of GI transit because the citation is a discussion of the effects of a loss-of-function mutation in TPH2, the rate-limiting biosynthetic enzyme for 5-HT in neurons of the ENS and the CNS.  Citation #10 is also relevant because it compares the effect of deletion of TPH1, which selectively eliminates 5-HT biosynthesis in EC cells and mast cells, to those of the deletion of TPH2, which selectively eliminates 5-HT biosynthesis in neurons.  The message of the two citations about 5-HT motility is that total GI transit time, as well as small and large intestinal propulsion, is dependent on the enteric neuronal store of 5-HT and not the epithelial 5-HT depot.  The authors may have been thinking about the results of the reports that they cite, but the text of their manuscript gives absolutely no indication of what they were thinking.  The English syntax of the writing is mysterious and thus lacks any meaning at all.  Worse, for a paper about 5-HT in the bowel and the role of 5-HT in the gut, the authors present no discussion of where 5-HT is located within the gut or how it gains access to ICCs under any physiological or pathophysiological conditions.  The authors treat ICCs as pharmacological assays. They apply exogenous 5-HT or 5-HT receptor antagonists to the cells, without any consideration of the physiological meaning, if any, of their observations or whether ICCs ever respond to endogenous 5-HT.  Do the authors have any evidence (none exists in the literature) that enteric serotonergic neurons synapse on ICCs?  The authors discuss the theory that ICCs are intermediates between cholinergic and nitregic neurons and smooth muscle.  Are they implying that a similar mechanism exists for 5-HT?  Do the authors think that 5-HT released within the myenteric plexus overflows from within enteric ganglia to reach ICCs, which are extraganglionic?  Alternatively, do the authors believe that 5-HT from the mucosal epithelium somehow flows through the lamina propria and submucosa to reach ICCs?  Given the expression of SERT within the GI epithelium and ENS, how would the spread of 5-HT though the layers of the gut be possible?  One would expect that uptake of 5-HT would prevent its wide dissemination.  In fact, that is what is considered to be essentially to confine the actions of 5-HT to its proximate physiological targets.  What then is the significance of the 5-HT receptors that ICCs express?  The authors never consider that question.  The manuscript is thus an exercise in pharmacology untethered to physiology.  

The conclusions on lines 356-362 are another example of incomprehensible English syntax.  The authors begin with the comment that:  “Several studies have revealed the important excitatory roles of 5-HT in GI motility; however, the mechanisms underlying this response have proven to be increasingly complex, especially in ICC.”  This sentence implies that the authors are going to provide some insight that would explain the complexity of these mechanisms.  Instead, the authors summarize their results one more time (repeating material written earlier): “In this study, we demonstrated that the effects of 5-HT on the pacemaker activity in small and colonic ICC are excitatory but with different patterns. ANO1, T-type Ca2+, 5-HT4 receptor and HCN channels are involved in 5-HT-induced effects, and MAPKs are partially involved in 5-HT-induced effects in colonic ICC.”  In other words, rather than explain the complexity of 5-HT effects on motility and ICC, the authors add to it.  There is nothing wrong with that, but it is out of place to complain about complexity and then add to it.

The authors need to consider the physiology of the bowel and the roles of 5-HT in it.  Alternatively, they might state that they are not at all concerned with either the actions of 5-HT in the gut or the functions of ICC. Instead, might claim to be anxious to gain insight into the particular subtypes of 5-HT receptor that ICC express for some other purpose that the authors might choose to propose.

Comments on the Quality of English Language

The paper needs to be re-written to consider the meaningfulness of individual sentences.

Author Response

Thank you for giving me the opportunity to submit a revised draft of our manuscript titled “5-Hydroxytryptamine Enhances the Pacemaker Activity of Interstitial Cells of Cajal in Mouse Colon” to International Journal of Molecular Sciences. We appreciate the time and effort that you and reviewers have dedicated to providing your valuable feedback on my manuscript. We have checked carefully with reviewers’ comments and corrected as their suggestions. For submitting our revisions, we have prepared the following documents and the changes made in the manuscript are marked with memo. Furthermore, we got English editing service from Editage company by English native speaker. We have attached the certification document.

Attached file is a point-by point response to the reviewers’ comments and concerns.

Reviewer 2 Report

Comments and Suggestions for Authors

Dear authors,

The purpose of your research was to demonstrate the important excitatory roles of 5-HT, especially in colonic ICC. To test your hypothesis, you used primary ICC cultures and observed the effect of several inhibitors in the presence and absence of 5-TH.

However, several questions make this article difficult to follow, and need clarification:

Results:

Line 63 – The authors should explain that they are using primary ICC cultures from colon and jejunum (?).

Line 64 – Since “ICC have a distinctive morphology that can be easily recognized in culture.”, can you include a figure with photos where ICC are identified?

Figure 2 - With ZD7288, CsCl, and DDA, the pacemaker activity of ICC decreased significantly before the application of 5-HT. Did this potent inhibitory effect interfere with the observed lack of response by 5-HT? Did you test other drugs that maintained its pacemaker effect in the presence of these inhibitors (positive control)?

Figure 3 a) How do you explain the different pacemaker effect (RMP) of Cisapride and prucalopride since they are both 5-HT4 agonists?

Figure 3 b) Why didn’t you use 5-HT receptor antagonist to clarify which receptor subtype is present in colonic and jejunal ICCs?

Figure 4 – What was the effect of SQ22536 in colonic ICC? And of ZD7288 and CsCl in jejunal ICCs?

Figure 5 – On small intestinal ICC, did you try other concentrations of HCN blockers (with 5-HT)?                                                                           

Figure 6  –You only tested the effect of ANO1 and T-type Ca2+ channel inhibitors on 5-HT-actions in colonic ICC. Did you also try these drugs on jejunal ICCs? Could the effect be different?                                                                           

Figure 7  – The tyrosine kinases inhibitor blocked the pacemaker activity of colonic ICC, but did not affect 5-HT-induced effects.

Line 222 – ICC =  intracellular carcinoma??

Discussion

Line 225 – 5-HT induced depolarization of membrane potentials and increased the frequency of pacemaker activity in colonic and jejunal ICC. How do you interpret the effect of all other inhibitors, regarding 5-HT effect? Summarize the differences between colonic and jejunal ICCs.

Line 277 - “the presence of ANO1 or T-type Ca2+ channel agonists, the pace-maker activity ceased in colonic ICC, which blocked the 5-HT-induced effects.” Did you use agonists or inhibitors?

Materials and Methods

Regarding the use of animals, please indicate:

a) the total number of animals used

b) how were animals sacrificed

c) line 318 – “The colon and small intestine were removed from the duodenum to the rectum and 318 the middle portion was used.” So, you used the middle colon and the jejunum?

Line 344 – did you record pacemaker potentials in the stomach?

Conclusions

. The conclusions should be an interpretation of the results, and not only a description of the data.

. It would help to have a final scheme, to visualize sequential 5-HT receptors signal transduction, that leads to an increase in ICCs pacemaker activity.

Thank you for presenting your work and I hope these suggestions are helpful.

I wish you luck and success.

Comments on the Quality of English Language

Line 18 - “With various agonists of 5-HT receptors, 5-HT4 receptor has a functional role for 5-HT in colonic ICC.” – Explain and rephrase

Line 25 - “Therefore, 5-HT has excitatory effects on pacemaker activity in small intestinal and colonic ICC but the patterns vary.” The word “intestinal” is missing.

Line 49 – “Therefore, the ICC can be a target in the regulation of GI motility and understanding the role of 5-HT in colonic ICC is essential for understanding the regulation of GI motility by the ICC.” Rephrase

Author Response

(The authors gave the same response as above.)

Round 2

Reviewer 2 Report

Comments and Suggestions for Authors

-

Comments on the Quality of English Language

-